# Fair Treatment and Job Satisfaction: A Multilevel Analysis of Employment Transition

**DOI:** 10.3390/bs15111524

**Published:** 2025-11-09

**Authors:** Hyunmin Cho, Kyujun Cho, Heungjun Jung

**Affiliations:** 1Department of Business Administration, Hanyang University, Seoul 04763, Republic of Korea; hmcho85@naver.com; 2Department of Business Administration, Seoul National University of Science and Technology, Seoul 01811, Republic of Korea; 0012hi@kli.re.kr

**Keywords:** fair treatment, job satisfaction, government support, employment transition, upward mobility

## Abstract

Drawing on organizational justice theory, this study examines how workers’ perceptions of fair treatment influence job satisfaction following the transition from temporary agency employment at subcontracting firms to regular employment with client firms. A multilevel analysis was conducted to simultaneously assess individual- and organizational-level effects. Data were collected through a survey of Korean public organizations that had implemented regular employment transitions, yielding a final sample of 966 employees nested within 116 institutions. At the individual level, perceived fair treatment after regularization was positively associated with job satisfaction. At the organizational level, systematic human resource management practices enhanced employees’ perceptions of fair treatment, while government support during the transition process—including the provision of clear guidelines and professional consulting—moderated the relationship between perceived fair treatment and job satisfaction. These findings contribute to a deeper understanding of how fair treatment perceptions shape employee attitudes following employment regularization and highlight the role of organizational human resource practices and government involvement in fostering positive outcomes during labor market transitions.

## 1. Introduction

Intensifying competition among firms, the globalization of neoliberalism, and technological advancements have weakened formal employment relationships, giving rise to various forms of precarious work, including temporary, contractual, and dependent labor arrangements ([44]). Employment casualization—such as a shift from full-time to contingent work—is inevitably accompanied by job insecurity, which can negatively affect both individual work attitudes and organizational performance ([48]). Conversely, in a labor market where atypical jobs are expanding more rapidly than traditional ones, transitioning from contingent to permanent employment can provide meaningful opportunities for individuals and enhance organizational efficiency by promoting long-term skill development ([43]). In particular, transitioning from employment at a subcontractor to a client firm offers workers the opportunity to attain more stable and favorable job conditions. Such transition opportunities can substantially enhance employees’ job attitudes following regularization, underscoring the strategic importance of upward mobility initiatives as a core component of effective human resource management.

While much research has focused on the precariousness of contingent work ([17]) and its causes, other studies have examined job attitudes in workplaces comprising both regular and non-regular employees ([50]; [56]). Existing research warns that the excessive use of non-standard employment negatively affects organizational effectiveness ([2]) and contributes to antagonistic job attitudes ([2]). This is linked to the finding that the relative deprivation experienced by non-standard workers may adversely impact job attitudes ([43]). Despite growing interest in employment transitions, prior studies have yet to clearly illuminate how workers perceive their own attitudes or how those attitudes evolve following conversion to permanent employment. To address this gap, the present study examines temporary agency workers’ perceptions of fairness and job satisfaction following their conversion to regular employment. Grounded in organizational justice theory, it is hypothesized that perceived fairness during the conversion process positively influences job satisfaction after regularization. Nevertheless, the interpretation of the findings warrants caution, as the analysis was based on perceptions of fair treatment reported by individuals who had already completed their transition to regular employment. This methodological consideration is acknowledged as a limitation of the study.

Furthermore, and more importantly, this study investigates whether a reciprocal relationship between individuals and organizations can be established following the transition from temporary to permanent employment. While prior research has primarily focused on the job attitudes of non-regular employees at the individual level ([74]), this study examines the organizational-level impact on employees’ psychological attitudes through multilevel analysis. Specifically, it analyzes the effects of human resource (HR) practices and government support policies on perceptions of fairness, suggesting that organizational-level policies may directly shape individual attitudes. The multi-level analysis indicates that organization-level HR practices positively influence employees’ perceptions of fair treatment, and that government support for full-time employment, including providing guidelines during the transition process and advising on debating agendas, moderates the relationship between perceived fair treatment and job satisfaction.

This study offers several important implications. First, organizations employing both regular and non-regular workers should actively promote and implement upward mobility opportunities. Notably, converting non-regular workers—particularly those seeking greater job stability—into regular employment can lead to significant improvements in attitudes and behaviors, potentially outweighing the perceived benefits of labor cost savings. Second, upward mobility should be supported by systemic HR practices. Organizational-level HR policies related to transitions into regular employment—such as structured job assignments and promotion pathways—can enhance perceptions of fair treatment. Third, external resources can create synergies during the employment regularization process. Our findings indicate that government support in public organizations strengthens the relationship between perceptions of fairness and job satisfaction.

## 2. Korean Context

This study employs a quantitative design and uses the Korean public sector as a contextual case to highlight its distinctive institutional characteristics. Although Korea’s public sector is government-operated, it has historically maintained a high proportion of non-regular workers and has undertaken extensive efforts to reduce such employment. The prevalence of non-regular work in Korea can be traced to the rapid adoption of neoliberal labor market reforms by private companies following the 1997 International Monetary Fund (IMF) bailout ([47]). From the early 2000s, neoliberal principles expanded beyond the private sector to the public domain under the paradigm of new public management. In pursuit of efficiency, the Korean government privatized public enterprises (e.g., telecommunications) and outsourced public sector labor, which resulted in the proportion of non-regular employees in the public sector reaching approximately 20% by 2006 ([54]). Most non-regular employees at that time were fixed-term workers, earning on average only about 50% of the wages of regular employees and facing considerable employment insecurity ([62]).

To address these inequities, the government enacted the Act on the Protection of Fixed-Term and Part-Time Workers in 2006. The legislation sought to safeguard the rights and interests of non-regular employees by mandating that fixed-term and part-time workers employed for two years or longer be converted to regular status. Since the law’s implementation in 2007, the government has steadily facilitated the regularization of fixed-term non-regular employees. However, by 2017, the overall proportion of non-regular workers in the public sector had not significantly declined. This stagnation largely resulted from the outsourcing of government tasks, which increased the number of subcontracted workers. While the number of fixed-term non-regular employees decreased, outsourcing generated a parallel rise in private contract workers.

In 2017, the Korean government launched stronger policy interventions to further reduce non-regular employment in the public sector. These initiatives expanded regularization efforts to include not only fixed-term employees but also subcontracted and dispatched workers performing ongoing and continuous work. Distinct from earlier policies, which were largely government-driven, the Moon Jae-in administration established a regularization committee to incorporate labor union participation and external expert consultation into the policy process. The government allocated a dedicated budget, formulated detailed implementation guidelines, and organized consulting teams to support public institutions in executing conversion plans.

These measures resulted in a systematic, nationwide regularization initiative that was implemented across approximately 900 public institutions between 2017 and 2022. During this period, the government provided standardized guidelines, monitored institutional compliance, and facilitated the transition of approximately 200,000 non-regular workers to regular employment. Among these, about 42% were fixed-term employees, while 58% were subcontracted or dispatched workers. Consequently, the proportion of non-regular employees in the public sector declined markedly—from 16.7% in 2017 to 5.5% in 2022 ([63]).

Building on this institutional setting, the present study examines the relationship between perceptions of fair treatment following regularization and job satisfaction, while also investigating the moderating role of perceived government support and HR policies in shaping these outcomes.

## 3. Theory Development and Hypotheses

### 3.1. Organizational Justice Theory and Employment Transition

According to Organizational Justice Theory, organizational justice is determined by the extent to which individuals perceive that they are treated fairly by their employing organization ([18]; [3]). Organizational justice is generally conceptualized as comprising three core dimensions: distributive justice, procedural justice, and interactional justice ([5]; [64]; [60]). Distributive justice refers to the perceived fairness of rewards, outcomes, and performance evaluations received from the organization ([11]). Procedural justice reflects the degree of fairness perceived in the processes by which rewards or outcomes are determined ([23]). Finally, interactional justice denotes the fairness of interpersonal treatment experienced during the decision-making process ([5]; [78]). Recent discussions suggest that organizations pursue fairness as a means of minimizing psychological inequality ([34]; [41]).

When individuals perceive justice in organizational relationships, they are more likely to voice challenging opinions, which in turn enhances work attitudes such as job engagement, conscientiousness, courtesy, and altruistic behavior ([26]; [1]; [86]). [12] ([12]) reported that higher levels of justice perception positively affect team performance while reducing absenteeism. Similarly, [14] ([14]) found that employees who experienced higher levels of justice reported greater psychological well-being compared to those who perceived lower levels of justice. Perceptions of justice are also associated with reduced turnover intentions ([27]; [10]), increased trust in the organization ([70]), and higher participation in organizational citizenship behaviors ([8]).

Furthermore, the degree of satisfaction employees derive from HRM (Human Resource Management) practices is directly linked to their perceptions of organizational justice ([21]). [24] ([24]) empirically confirmed a positive relationship between HRM practices and organizational justice. [35] ([35]) found that perceptions of fairness in the recruitment process were positively associated with applicants’ acceptance decisions. Organizational justice perceptions have also been shown to mediate the relationship between HRM practices and both OCB-I (Organizational Citizenship Behavior–Individual) and OCB-O (Organizational Citizenship Behavior-Organization) ([81]). In addition, [69] ([69]) provided empirical evidence that perceptions of procedural justice are related to organizational commitment through the mediating role of perceived organizational support (POS).

Social exchange theory is widely used in research on organizational fairness and in theorizing the relationship between individuals and organizations in the workplace ([13]). Social exchange theory (SET) is considered one of the most influential conceptual paradigms for understanding workplace behavior ([13]). It is premised on the principle of reciprocity. [57] ([57]) equate social exchange with reciprocity, which can be initiated by an organization’s treatment of its employees with the expectation that the treatment will be reciprocated. Exchange resources can be tangible and economic—such as money or goods—but social resources, including status and information, are also important. This is because social exchange theory emphasizes socioemotional outcomes, such as social esteem and self-fulfillment, alongside economic outcomes like compensation ([29]). These principles form a critical foundation for predicting employee behavior following an employment transition.

Prior research indicates that organizational justice theory and social exchange theory are useful in explaining employee attitudes and performance not only in traditional employment relationships but also in evolving employment contexts where occupational status changes ([9]). The transition to permanent employment following a fixed-term or temporary position is associated with occupational upward mobility, which shapes the relationship between employees and organizations ([22]).

### 3.2. Perceived Fair Treatment and Job Satisfaction

Fair treatment is often used interchangeably with justice but can be broadly conceptualized as respectful and equitable behavior ([55]). It is closely related to interactional justice, as employees perceive fairness or unfairness in their interactions with organizational decision-makers ([5]). [59] ([59]) defined fair treatment as treating everyone equally, regardless of personal preferences or private relationships. Fair treatment serves as a normative principle that is particularly important in employment relationships within public organizations. When organizations treat employees fairly and without bias, employees are more likely to comply with organizational directives and exhibit positive changes in their attitudes. For instance, prior studies have shown that fair treatment reduces turnover intentions ([36]), lowers stress levels ([80]), and enhances organizational commitment ([65]).

This study posits that transitioning to permanent employment may heighten perceptions of organizational fairness, thereby improving job satisfaction. Numerous studies have demonstrated that fair treatment by organizations elicits positive employee responses ([16]; [52]). When employees believe they are treated fairly, they tend to perceive organizational decisions as just and are more likely to trust them. This not only increases acceptance of those decisions but also provides psychological stability, leading to higher levels of job satisfaction ([76]). Fair treatment fulfills employee expectations by demonstrating respect and impartiality. Accordingly, previous research has consistently reported a positive relationship between perceived fair treatment and job satisfaction ([68]). [51] ([51]) further showed that fair interactions, in the context of Turkish public and private bank employees, can reduce negative emotions toward the organization and increase job satisfaction.

**Hypothesis** **1.***At the individual level, perceived fair treatment is positively associated with job satisfaction*.

### 3.3. HR Practices and Perceived Fair Treatment

This study investigates how human resource (HR) practices shape perceptions of fairness among workers who have undergone employment transitions. Drawing on the motivational processes outlined in social exchange theory ([6]), the research posits that HR practices function as mechanisms through which employees interpret organizational treatment and intent. Through HR practices implemented by the organization and enacted by managers, employees form judgments about how they are valued and supported. These practices also serve as signals of the organization’s commitment, shaping employees’ understanding of the employment relationship. In line with signaling theory, HR practices convey implicit messages about expectations of mutual and reciprocal exchange ([30]).

When HR practices—such as employee recruitment, training, performance evaluation, and compensation systems—adhere to ethical standards and follow fair procedures, employees are more likely to perceive that they are being treated fairly by the organization ([75]). Recent research has also emphasized the importance of individual justice sensitivity in the context of fair HR practices guided by clear rules and procedures ([32]). When practices related to rewards, promotion criteria, and task distribution are based on well-established principles, employees are less likely to feel disadvantaged. Conversely, when no transparent criteria exist for rewards, training, or task assignments, employees may perceive such practices as unfair.

Workers who transition from non-regular to regular positions may interpret previously unfamiliar “soft” and high-commitment human resource (HR) practices as signals of fair treatment within the organization. As temporary agency workers, employees typically lack access to promotion systems, differentiated compensation based on role, and career development training. In contrast, regular employees are subject to structured performance evaluations, promotion opportunities, and formal compensation systems, which can enhance motivation and foster a sense of organizational fairness. Prior research has empirically shown that reciprocity-based HR practices promote positive employee attitudes ([45]). Building on this perspective, the implementation of systematic HR practices following a worker’s transition from a temporary agency role to a regular position within the client firm is likely to shape perceptions of fair treatment.

**Hypothesis** **2.***At the organizational level, HR practices are positively related to perceived fair treatment*.

### 3.4. Moderation Effects of Perceived Government Support

Perceived government support at the organizational level may moderate the relationship between perceived fair treatment and job satisfaction at the individual level. Previous studies have consistently demonstrated that social resources from colleagues and supervisors are positively associated with job attitudes ([33]). In the context of public sector employment, government support serves as a critical external social resource ([51]). Specifically, governments facilitate the transition of contingent workers into regular employment by enacting supportive policies and regulations, offering financial mechanisms such as subsidies, tax incentives, and direct funding, and by overseeing the implementation of these measures to ensure equitable treatment and prevent misuse.

This study hypothesizes that government support moderates the relationship between perceived fair treatment and job satisfaction. Since employment transitions in public institutions have been driven by government initiatives, institutions that perceive stronger governmental support are more likely to implement systematic preparation and fair procedures during the conversion process. Under such conditions, employees who transition to regular positions are expected to exhibit a stronger positive association between perceived fair treatment and job satisfaction. Third-party mediation may be beneficial, as the interests of workers and employers often diverge during the employment transition process. When government support is robust and consistent, specific regulations related to employment transitions are likely to be in place, and the necessary budget will be secured. In such cases, worker representatives, management, and experts can collaboratively establish transition plans that minimize discrimination against existing regular employees.

Conversely, when government support is weak or inconsistent, public institutions may fail to reduce discriminatory practices during the employment transition process. In such instances, the relationship between perceived fair treatment and job satisfaction may be diminished. If the government does not provide clear guidelines or sufficient oversight, criteria for employment transition may remain vague, and labor-management conflicts may persist. Under these conditions, the positive impact of perceived fair treatment on job satisfaction is likely to be weakened. Therefore, the degree of government support may either strengthen or weaken the effect of perceived fair treatment on job satisfaction.

**Hypothesis** **3.***Perceived government support at the organizational level moderates the relationship between perceived fair treatment and job satisfaction, such that the relationship is stronger when perceived government support is high*.

Figure 1 shows the theoretical framework synthesizing the hypotheses.

## 4. Methods

### 4.1. Research Setting, Sample, and Procedure

To obtain a sample aligned with the study variables, data were collected from South Korean public organizations that had transitioned irregular employees into regular positions. Stable employment relationships tend to be more pronounced in the public sector, primarily for two reasons. First, job security is typically greater in the public sector than in the private sector, which fosters long-term employment relationships. Second, public- sector employers are expected to model exemplary labor practices, resulting in increased legal responsibility and accountability. For these reasons, prior studies have applied social exchange theory to explain employee attitudes and performance in the public sector ([79]).

Organizational-level measures—human resource (HR) practices and perceived government support—were completed by HR personnel in each institution. Individual-level measures were completed by employees who had transitioned from non-regular to regular employment. Of the 853 public organizations in South Korea that experienced employment transitions, 846 were contacted for participation, excluding those that had merged or lacked available contact information.

Questionnaires were mailed to HR managers responsible for the regularization process. This survey was part of a broader research project investigating full-time employment transitions in Korean public institutions. The mailing included a cover letter outlining the purpose of the study, voluntary participation, and confidentiality, as well as a return address for completed surveys. Data collection occurred over a three-week period, resulting in 142 completed surveys and a response rate of 16.8%.

In organizations where HR responses were secured, researchers conducted follow-up, in-person surveys with cooperation from the organizations. Questionnaires were randomly distributed to workers who had experienced employment transitions. Between 10 and 30 questionnaires were distributed per organization, depending on its size. A total of 2560 questionnaires were distributed, with 1372 returned, yielding a response rate of 53%.

Following recommendations for multilevel analysis, organizations with fewer than three employee responses were excluded ([58]; [82]). After removing incomplete responses and unmatched HR manager–employee dyads, the final sample consisted of 116 HR managers and 966 individual employees, with an average of eight participants per organization.

Descriptive statistics for the sample indicated that organizations employed, on average, 1254 individuals. The average age of participants was 42 years, and 63% identified as female. Regarding educational attainment, 1.7% of participants held a middle school diploma or lower, 22.6% had a high school diploma, 66.6% held a college degree, and 9.2% had a graduate degree or higher. At the individual level, 71.9% of employees were members of a labor union. At the organizational level, 59% reported the presence of a labor union.

### 4.2. Measures

Following [39]’s ([39]) iterative back-translation method, all survey items were translated into Korean.

#### 4.2.1. Perceived Fair Treatment

Perceived fair treatment was measured to be to the extent to which employees who experienced a transition from irregular to regular employment perceived being treated fairly by their organization. The measure employed a five-item scale developed by [71] ([71]). This measurement tool functions as a perceptual proxy for overall process fairness, integrating key aspects of distributive justice and procedural justice experienced by workers within specific personnel-related interactions. Responses were rated on a 5-point Likert scale ranging from 1 (strongly disagree) to 5 (strongly agree). Sample items included: after being converted to a regular position, (1) “I am treated fairly in my job assignment,” (2) “I am treated fairly in my pay,” (3) “I am treated fairly in my discipline,” (4) “I am treated fairly in my career advancement,” and (5) “I am treated fairly in my training.” Scores across the five items were averaged to form a composite index of fair treatment. The scale demonstrated strong internal consistency (Cronbach’s α = 0.85).

#### 4.2.2. Job Satisfaction

Job satisfaction was assessed using the five-item scale originally developed by [40] ([40]) and utilized in recent research ([77]). Participants responded on a 5-point scale ranging from 1 (not at all) to 5 (extremely). Items included: after being converted to a regular position, (1) “I am satisfied with my job,” (2) “my colleagues,” (3) “the supervisor,” (4) “the promotion,” and (5) “my income.” Internal reliability for the scale was acceptable (Cronbach’s α = 0.83).

#### 4.2.3. HR Practice

HR practices were measured based on three categories identified by [15] ([15]): staffing, appraisal, and promotion—selected for their relevance to this study’s context. Although some studies suggest that HR practices reported by employees may better predict attitudes ([67]), such measures can introduce common method bias. Therefore, HR practices were assessed at the organizational level. HR managers were asked to indicate whether each practice applied to workers who had transitioned to regular employment in their organization. Each practice was coded as 1 (present) or 0 (absent), and total scores ranged from 0 to 3.

#### 4.2.4. Perceived Government Support

Perceived government support was measured using three items reflecting HR managers’ perceptions of governmental support in the conversion of irregular workers to regular positions. The scale was adapted from [53] ([53]) and [42] ([42]). Items included: (1) “The experts sent by the government helped our organization transform irregular employees into regular employees,” (2) “The government guidelines were helpful during our organization’s conversion process,” and (3) “The government encouraged our organization to proceed with employment transitions.” Responses were rated on a 5-point scale ranging from 1 (not at all) to 5 (extremely). Internal consistency was acceptable (Cronbach’s α = 0.78).

#### 4.2.5. Control Variables

Employees’ demographic variables could account for variance in job satisfaction and public service performance ([61]). At the individual level, we controlled for sex, age, education level, and union membership status ([73]). At the organizational level, we controlled organizational size (measured by total number of employees and log-transformed), type of organization (1 = central or local government/public enterprise; 2 = subsidiary of public enterprises; 3 = contracted public organization), and the presence of a labor union ([19]).

### 4.3. Data Analyses

Prior to hypothesis testing, we evaluated our proposed four-factor measurement model using multilevel confirmatory factor analysis (MCFA) (Table 1). The baseline model, which distinguished between employee-reported factors (perceived fair treatment and job satisfaction) and manager-reported factors (human resource practices and perceived government support), demonstrated acceptable overall model fit (χ^2^ (36) = 206.55; **C**omparative **F**it **I**ndex (CFI) = 0.91; **T**ucker-*L*ewis **I**ndex (TLI) = 0.90). Although the **R**oot **M**ean **S**quare **E**rror of **A**pproximation (RMSEA) value of 0.07 and the within-group **S**tandardized **R**oot **M**ean Square **R**esidual (SRMR) value of 0.09 slightly exceeded conventional cutoff thresholds ([38]), it is important to note that these criteria were developed for single-level analyses and may be overly restrictive in multilevel contexts ([49]; [84]).

Several pieces of evidence support the adequacy of the proposed measurement model. First, the between-group SRMR value of 0.03 indicated excellent model fit at the organizational level. Second, all comparative fit indices (CFI and TLI) exceeded the recommended threshold of 0.90. Most importantly, the four-factor model showed significantly superior fit compared to a theoretically plausible two-factor model that combined all employee-reported items and all manager-reported items, respectively (Δχ^2^ (1) = 262.48, *p* < 0.001). The poor fit of the alternative model (CFI = 0.78; RMSEA = 0.11; SRMR within = 0.13) provided strong evidence for the discriminant validity of our proposed factor structure.

Following contemporary recommendations to assess model fit holistically rather than relying on single indices ([72]), and considering both comparative fit and theoretical coherence, we concluded that the four-factor model was suitable for testing our hypotheses. This conclusion was further supported by the model’s ability to maintain conceptual clarity between constructs while accounting for the hierarchical structure of the data.

#### Data Analytical Strategy

To address the limitations of aggregation and disaggregation biases inherent in multilevel data, we employed hierarchical linear modeling (HLM) and followed the mediation testing procedures outlined by [4] ([4]). HLM enables the simultaneous analysis of nested data structures, allowing for the examination of relationships between variables at multiple levels while accounting for their distinct sources of variance ([37]).

Additionally, HLM is well-suited for testing cross-level interaction effects, in which group-level predictors influence individual-level outcomes ([37]). In this study, we constructed a series of hierarchical models corresponding to different levels of analysis. Specifically, we calculated hierarchical regression equations for each individual at Level 1 and then used the resulting intercepts and slopes as dependent variables in Level 2 models. A significant parameter estimates at Level 1 indicated an individual-level effect, whereas a significant Level 2 predictor of the Level 1 intercept or slope indicated a group-level effect.

## 5. Results

### 5.1. Respondent Characteristics

Table 2 presents the means, standard deviations, and correlations among the study variables, offering a comprehensive overview of our sample’s key characteristics. Our sample, comprising 996 regularly converted employees across 116 public organizations, exhibits the following features. Regarding gender distribution, 611 participants (63.3%) were female, and 355 participants (36.7%) were male. While this differs from the overall South Korean public sector’s gender distribution (men 51.5%, women 48.5%), it reflects the higher proportion of women typically found among non-regular public sector employees. Education levels were distributed as follows: 16 participants (1.66%) had completed middle school or less, 218 participants (22.6%) high school or less, 643 participants (66.6%) college or less, and 89 participants (9.21%) had a university degree or higher. The average age of participants was 42.1 years (SD = 10.44), closely aligned with the predominant age group in the public sector (40s). The average job tenure was 7.09 years (**S**tandard **D**eviation (SD) = 4.35), notably shorter than the overall public sector average of 11.3 years. This shorter tenure is an expected characteristic given our specific focus on regularly converted employees. At the individual level, 684 participants (70.8%) were union members, while 282 participants (29.2%) were non-members. At the organizational level, among the 116 organizations, 69 organizations (59.5%) had a labor union, whereas 47 organizations (40.5%) did not. For the types of public organizations, 99 organizations (85.3%) were Public Institutions, and 17 organizations (14.7%) were Local Public Enterprises.

### 5.2. Hypothesis Testing

Table 3 presents the results of the hierarchical linear modeling (HLM) analysis. At the individual level, the study hypothesized that perceived fair treatment would positively influence job satisfaction. The results from Model 1 indicated that perceived fair treatment had a significant positive effect on job satisfaction (β = 0.57, *p* < 0.01) and improved model fit compared to the null model, supporting Hypothesis 1.

At the cross-level, the study predicted that human resource (HR) practices would positively influence perceptions of fair treatment. Results from Model 2 demonstrated a significant positive relationship between HR practices and perceived fair treatment (β = 0.08, *p* < 0.05), along with improved model fit over the null model. Thus, Hypothesis 2 was supported.

Perceived government support was expected to moderate the positive relationship between perceived fair treatment and job satisfaction. Model 3 tested the effects of both perceived fair treatment and perceived government support on job satisfaction. The effect of perceived fair treatment remained significant (β = 0.57, *p* < 0.01), while perceived government support was not significant. Model 4 introduced the interaction term between perceived fair treatment and perceived government support to test the moderation effect. The interaction term was significant (β = 0.06, *p* < 0.01), indicating a positive moderating effect. Model fit also improved, supporting Hypothesis 3.

While our primary analysis employed a composite measure for general job satisfaction, we acknowledge the importance of disaggregating this construction to understand the nuanced impact of perceived fair treatment on specific aspects of employees’ work experience. This approach, which allows for a more granular examination of psychological mechanisms (e.g., [7]), was further prompted by valuable feedback. Consequently, we conducted an additional series of multilevel regression analyses, using fair treatment as the independent variable and each of the five job satisfaction facets (satisfaction with job itself, colleagues, supervisor, promotion, and income) as distinct dependent variables, while maintaining the same control variables as our main models. According to the results, fair treatment exhibits a statistically significant and strong positive influence on all five facets of job satisfaction (all *p* < 0.01). Furthermore, perceived fair treatment had the strongest effect on satisfaction with income (β = 0.68) and satisfaction with colleagues (β = 0.64). These results confirm that perceived fair treatment is a pervasive predictor across the various dimensions of job satisfaction in our sample. The results of this exploratory analysis are summarized in Table 4.

To further investigate the complex role of unions in shaping employee attitudes, particularly their influence on the crucial link between perceived fair treatment and job satisfaction, we conducted additional multilevel moderation analyses (Table 5). This approach allowed us to differentiate between the moderating impacts of individual-level union membership and organizational-level existence of union ([31]). The results of these two moderation models are presented in Table 2. Model 1 reveals a statistically significant negative interaction effect between fair treatment and individual-level union membership (β = −0.11, *p* < 0.05). Individual-level union membership can attenuate the positive impact of fair treatment on job satisfaction by fostering a heightened critical awareness of workplace conditions, redirecting focus towards collective grievances rather than personal gains. These findings may be understood as reflecting the role of labor unions in heightening workers’ critical awareness of workplace conditions within unionized organizations. This pattern aligns with the concept of the “paradox of union dissatisfaction” ([25]), which suggests that employees in unionized settings may report greater dissatisfaction despite objectively better working conditions compared with those in non-unionized firms. In contrast to the individual-level findings, Model 2 shows that the interaction between fair treatment and organizational-level union presence was not statistically significant (β = −0.05, *p* > 0.05).

Figure 2 presents the Johnson–Neyman plot illustrating the moderating effect of perceived government support on the relationship between perceived fair treatment and job satisfaction. The slope of perceived fair treatment’s effect on job satisfaction is significantly positive (β = 0.60, *p* < 0.001) and increases as perceived government support levels rise. Specifically, the interaction term between perceived fair treatment and perceived government support is significant (β = 0.06, *p* = 0.008), indicating that the positive impact of perceived fair treatment on job satisfaction is stronger at higher levels of perceived government support. Across the observed range of perceived government support, this relationship remains statistically significant, demonstrating that employees’ perception of fairness has a more pronounced effect on their job satisfaction in organizations with greater perceived government support.

## 6. Discussion

This study investigates the relationship between perceived fairness and job satisfaction among workers who have transitioned from contingent to permanent employment within South Korea’s public sector, where government-led initiatives actively promote the regularization of non-standard workers. While contingent labor offers employers advantages such as workforce flexibility and reduced labor costs, prior research has argued that triangular employment relationships—such as those involving temporary agency workers—can significantly reduce employers’ legal obligations ([83]). Despite these efficiencies, contingent workers often feel relatively disenfranchised. This study addresses this discrepancy by proposing that transitioning to permanent employment can alter workers’ fairness perceptions, thereby fostering more favorable job attitudes.

This study provides several theoretical implications. First, it contributes to the literature by examining perceptions of fair treatment and corresponding employee responses after the employment transition process. While previous research has largely conceptualized fair treatment in terms of interpersonal respect and supervisor-subordinate relationships, this study focuses on fair treatment in an organizational context—specifically, the conversion of contingent workers to standard employment. Recent studies have emphasized the importance of organizational context in shaping fairness perceptions ([56]; [46]; [85]). Organizational changes, such as employment transitions, may alter employees’ fairness perceptions and thereby influence their behaviors. This study extends prior work by demonstrating that fair treatment of newly regularized employees, as part of broader organizational policy, can positively affect job attitudes such as job satisfaction.

Second, the findings suggest that organizational-level signals, such as human resource (HR) practices, significantly influence individuals’ perceptions of fair treatment. Through multilevel analysis, it was confirmed that HR practices—including systematic work assignments, compensation structures, and training—positively impact individual-level fairness perceptions. These findings build upon the work of [28] ([28]), who explored the relationship between HR practices and employee outcomes from a social exchange theory perspective. While their study focused on “soft” HR practices as perceived by individuals, the current study tested the effects of objectively implemented HR practices at the organizational level on individual psychological outcomes.

Although a primary role of HR management is to ensure fair opportunities—such as designing performance-based compensation systems—contingent workers, particularly those employed on contracts or in temporary roles, are often excluded from systematic personnel management. In South Korea, contingent workers are rarely unionized, often earn minimum wages, and have limited access to training or career development programs. Consequently, their job satisfaction tends to be lower than that of regular employees ([66]). However, once they transition into standard employment, these individuals benefit from structured compensation systems and gain access to development opportunities such as training and promotion pathways. As a result, they are more likely to perceive organizational practices as fair and consistent, enhancing their sense of equitable treatment.

Third, this study confirms that government support can function as a form of social resource. Specifically, the relationship between perceived fair treatment and job satisfaction was found to vary based on perceived levels of government support. While prior research has consistently demonstrated that organizational and peer support can act as important social resources ([20]), this study extends those findings by showing that, in the public sector, government support can similarly serve this function. When public organizations perceive government support as strong and effective, the positive relationship between perceived fair treatment and job satisfaction is strengthened. These results imply that government initiatives not only reduce the financial and administrative burdens of employment transitions for public institutions but also reinforce employees’ positive job attitudes by affirming their perceived fair treatment post-transition.

This study offers important insights into how perceptions of fair treatment shape job satisfaction following the transition from temporary to regular employment. However, several limitations should be noted. First, perceptions of fairness were measured only after the transition process. A more rigorous design would assess fair treatment both before and after regularization to capture changes over time more accurately. Second, the study focused exclusively on employees who transitioned to regular status. Including a comparison group of workers who remained temporary agency employees would enable a deeper understanding of how employment type shapes perceived fairness treatment and job attitudes. Future research should adopt longitudinal and comparative designs to address these limitations and provide a more dynamic view of fairness during employment transitions. Third, the organizational-level response rate was 16.8%, which may constrain the generalizability of the findings. This relatively modest rate reflects the challenges of the study’s multilevel design, which required linking individual responses to their affiliated institutions. Nevertheless, obtaining 142 completed surveys from public organizations constitutes a valuable organization–individual dataset, particularly given the sensitive nature of the research topic.

## Figures and Tables

**Figure 1 behavsci-15-01524-f001:**
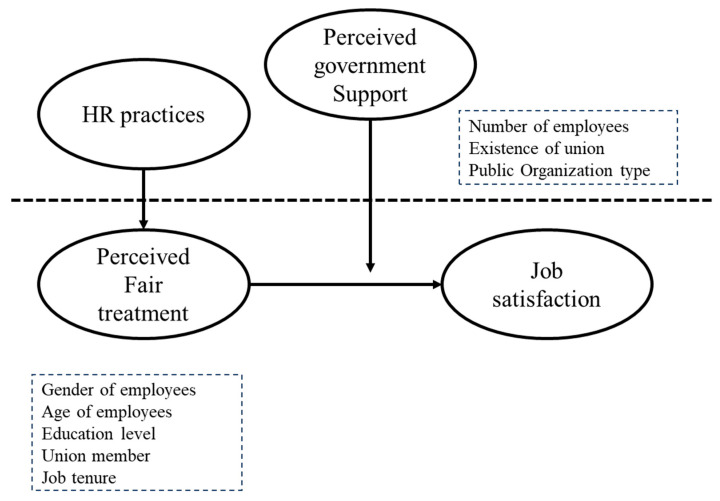
Theoretical model.

**Figure 2 behavsci-15-01524-f002:**
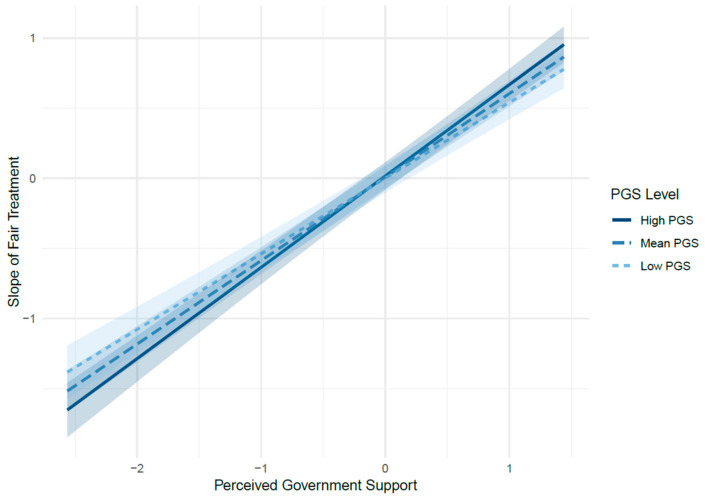
Johnson–Neyman Plot for the Moderating Effect of Perceived Government Support on the Relationship between and Job Satisfaction. PGS: Perceived Government Support.

**Table 1 behavsci-15-01524-t001:** Confirmation factor analysis.

Model	χ^2^(df)	RMSEA	SRMR Within	SRMR Between	CFI	TLI
Model 1: 4 factors	206.55 (36)	0.07	0.09	0.03	0.91	0.90
Model 2: 2 factors	469.03 (37)	0.11	0.13	0.03	0.78	0.69

*Notes*. N = 966. Model 1: Base line four-factor model with perceived fair treatment, job satisfaction, human resource practices, perceived government support. Model 2: Two-factor model with perceived fair treatment, job satisfaction, and human resource practices, perceived government support. Abbreviations: CFI, comparative fit index; RMSEA, root mean square error of approximation; SRMR between, standardized root mean squared residual between department; SRMR within, standardized root mean squared residual within department.

**Table 2 behavsci-15-01524-t002:** Means, standard deviations, and correlations.

Variable	Mean	SD	1	2	3	4	5	6
Individual Level								
1. Age	42.1	10.44						
2. Sex	0.63	0.48	0.07 *					
3. Education level	2.83	0.6	−0.36 **	−0.09 **				
4. Union member	0.71	0.45	0.00	0.03	−0.01			
5. Job tenure	7.09	4.35	0.27 **	−0.11 **	−0.02	0.01		
6. Perceived fair treatment	3.56	0.82	0.11 **	−0.10 **	−0.09	0.01	0.07 *	
7. Job satisfaction	3.42	0.77	0.11 **	−0.05	−0.03	−0.05	0.05 *	0.64 **
Organizational Level								
1. Number of employees	1254.52	2645.85						
2. Existence of union	0.59	0.49	−0.25 **					
3. Public organization type	1.15	0.36	−0.09	−0.04				
4. HR practices	1.22	1.11	0.17	0.19 *	0.05			
5. Perceived government support	3.58	0.81	0.17	−0.42 *	−0.15	−0.17		

*Note*. Individual Level N = 966 Organizational Level N = 116; * *p* < 0.05, ** *p* < 0.01. SD: standard deviation; HR: human resource. Gender: 0 = male and 1 = female. Education: 1 = middle school diploma and below, 2 = high school diploma, 3 = bachelor’s degree and above. * *p* < 0.05; ** *p* < 0.01.

**Table 3 behavsci-15-01524-t003:** Results of the hierarchal linear modelling analysis.

	Job Satisfaction	Perceived Fair Treatment	Job Satisfaction	
	Model 1	Model 2	Model 3	Model 4
Intercept	0.01	0.11 *	0.01	0.01
Level 1—Employee				
Age	0.01 *	0.01	0.01 *	0.00 *
Sex	0.00	−0.15 *	0.00	0.01
Education	0.06	−0.07	0.06	0.06
Union member	−0.11 *	0	−0.11 *	−0.12 *
Job tenure	0.00	0.00	0.00	0.00
Perceived fair treatment	0.60 **		0.60 **	0.60 **
Job satisfaction		-		
Level 2—Manager				
Number of employees	−0.02	0.03	−0.02	−0.02
Existence of union	0.09	0.04	0.08	0.08
Public organization type	0.04	0.11	0.03	0.02
HR practices	-	0.06 *	-	-
Perceived Government support	-	-	0.01	0.01
Cross-level moderation				
Perceived fair treatment × Perceived government support	-	-	-	0.06 *
*Marginal R* ^2^	0.42	0.03	0.42	0.42
*Conditional R* ^2^	0.48	0.16	0.48	0.49

*Note*. Individual Level N = 966 Organizational Level N = 116; * *p* < 0.05, ** *p* < 0.01. HR: human resources; *R*^2^: squared correlation.

**Table 4 behavsci-15-01524-t004:** Direct Effects of perceived Fair Treatment on Job Satisfaction Dimensions.

Dependent Variable	Model 1	Model 2	Model 3	Model 4	Model 5
Job satisfaction	my job	colleagues	supervisor	promotion	income
Intercept	2.63 **	3.02 **	3.57 **	3.65 **	3.07 **
Perceived Fair treatment	0.54 **	0.64 **	0.55 **	0.55 **	0.68 **
(Std. Error)	−0.03	−0.04	−0.03	−0.03	−0.04
(t-value)	−16.59	−17.75	−16.8	−17.21	−17.63

*Note*. Individual Level N = 966 Organizational Level N = 116; ** *p* < 0.01. Level 1 (Individual−level controls): Sex, Age, Education, Job Tenure, Union Membership. Level 2 (Organizational−level controls): Number of Employees, Organizational Step, Existence of union.

**Table 5 behavsci-15-01524-t005:** Moderation Effects of Union Membership and Presence on Perceived Fair Treatment and Job Satisfaction.

Dependent Variable: Job Satisfaction	Model 1 (Individual-Level Union)	Model 2 (Organizational-Level Union)
**Fixed Effects**		
Intercept	0.01	0.01
Perceived fair treatment	0.59 **	0.59 **
Individual-Level Union Membership	−0.12 **	-
Organizational-Level Existence of union	-	0.09
**Interaction Term**		
Perceived fair treatment × Individual-Level Union Membership	−0.11 *	-
Perceived fair treatment × Organizational-Level Existence of union	-	−0.05

*Note*. Individual Level N = 966 Organizational Level N = 116; * *p* < 0.05, ** *p* < 0.01, Level 1 (Individual-level controls): Sex, Age, Education, Job Tenure. Level 2 (Organizational-level controls): Number of Employees, Organizational Step.

## Data Availability

The data presented in this study are available upon request from the corresponding author.

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
