# Peer review of "Fair Treatment and Job Satisfaction: A Multilevel Analysis of Employment Transition"

_behavsci, 2025, doi:10.3390/bs15111524_

Round 1

Reviewer 1 Report

Comments and Suggestions for Authors

In this manuscript, the authors present the results of a study aimed at analyzing the impact of perceived fair treatment on employees’ job satisfaction following transitions to full-time employment in South Koreans public sector organizations. The hypotheses, grounded in the Social Exchange Theory, are clear and sound, and the multilevel analysis is appropriate for examineìing both individual-level and organizational-level implications. I have only a few minor comments to further improve the overall quality of the manuscript:
-    Rows 122-123: “According to motivation theory, individuals strive to achieve goals when they perceive value in doing so.” Citation needed
-    Please check the manuscript for typos (e.g., the 3.2 Fair Treatment and Job Satisfaction section is in italics, same for 4.2 section “Measures”, rows 322 “factors (fair treatment and job satisfmanager reported factors” etc)
-    A Cronbach’s alpha of .85 is considered as strong internal consistency (Fair treatment scale) but a value of .83 is just acceptable (job satisfaction), please check
-    In Table 2, please report frequencies for ordinal and categorical variables (such as Education level and Union member) instead of means and SDs
-    Please add the low organizational-level response rate (16.8%) to the study’s limitations.

Author Response

Thank you for your review. I have attached the response document. 

Reviewer 2 Report

Comments and Suggestions for Authors

Thank you for the opportunity to review this research. Overall, the article is well presented and organized, and the established methodology is well evaluated.
I only have two questions:
1) It should be specified that this is a case study, since in other countries, it is very unlikely to find public servants who are not within the legal framework.
2) I don't see the variable of job tenure. I believe this variable is essential to understanding the phenomenon studied.

Author Response

(The authors gave the same response as above.)

Reviewer 3 Report

Comments and Suggestions for Authors

Summary: Using data on public organizations in Sout Korea the authors explore: a) whether workers’ perceived fair treatment (related to transitioning firm temporary to permanent employment) affects their job satisfaction and the extent to which government support mediates the link and b) whether organizations’ HR practices  impact workers’ perceived fairness of their treatment.

General assessment: The paper is easy to read; it is well structured and well-motivated. I have some concerns that I ask the authors to consider incorporating in the current paper (see a and b) or use in a separate work to complement the paper c): a) work on improving directness/clarity of what the paper is doing early on in the paper, b) measurement concern in terms of bundling satisfaction over promotion linked to transitioning together with other more broader dimensions of job satisfaction, and c) sample selection concern over absence of data on workers who did stayed at less attractive jobs

Detailed comments:

  1. Abstract discusses fair treatment linked to employment transition but never clearly defines what this transition is – does it mean a person has just been hired or has he/she been assigned a new job? Please clarify.  The first paragraph in the Introduction is clearer as it discusses transitioning between different contract types (temporary/fixed-term vs permanent employment).

  1. Abstract would also benefit if it were clearer as to what kind of government support is being assessed as having mediating effect. Is it in terms of money firms get or does the money go to workers? Is the support non-monetary?  If yes, what does that mean?  This is mentioned later on but would be helpful to know in the Abstract.

  1. Abstract mentions that the authors use data on public organizations in South Korea - does public mean publicly traded on the stock market or public sector? Bottom paragraph in section 2 (on context) clarifies hat this could be about public sector – but it would be useful to have this information early on.

  1. In Section 2 (top paragraph on page 3) the authors discuss how temp agency workers were converted to permanent – how? From what perspective? In terms of the benefits, they were entitled to or also in terms of length of their employment? Is it in terms of firing costs terms agencies faced?  Please clarify.

  1. In section 3.1., the authors discuss social exchange theory on reciprocity but do not end up testing implications of this theory as I understand the paper. Can they include it as a part of a framework that is examined empirically?  Is it only in the case the theory is valid that we should observe validity of hypotheses spelled out later on in the paper?

  1. Hypothesis 1: isn’t it about perception of whether treatment was fair rather than actual fairness? The same for Hypothesis 2 – HR practices at organizational level affect the workers’ perceived fairness of their treatment. I think it makes sense to be clear about this d44possible distinction between perception and actual.

  1. I would also be interested in the role of unions – do they have a mediating effect (is the effect larger or smaller compared to government’s)? This is something that could easily be checked it seems.

  1. In terms of the data: it would be of interest to have a sense/measure of perception of fair treatment more generally, in the absence of transitioning (for workers who stayed at temp jobs). Do they perceive to be treated fairly?  It is not clear to me.  Do the authors have such measure, and can it be used in the analysis?  In some way it is expected to observe higher satisfaction for those that got promoted so that the current paper mostly is about figuring out the magnitude of the link. 

  1. In terms of job satisfaction (4.2.2.) I think it would be worthwhile to allow for a distinct link to four measures of satisfaction (regarding job, colleagues, supervisor, and promotion). It seems the satisfaction measured along the latter dimension (promotion or transitioning) is most relevant.  Why bundle all these measures in a single measure?

Author Response

(The authors gave the same response as above.)

Reviewer 4 Report

Comments and Suggestions for Authors

The paper uses an employer-employee dataset of Korean governmental organisations and their employees to analyse how HR-practices affect fairness perceptions and job satisfaction. It focusses on employees who recently transitioned from unregular to regular employment. The paper addresses a relevant research question and the authors should be applauded to have collected such a great dataset. Nevertheless, I had several question marks about the research design when reading the paper. These question marks have also been described as limitation at the end but I think they severely limit the value added of the paper. The paper positions its contribution as understanding perceptions and attitudes for workers transitioning from unregular to regular employment but cannot address the research question with data of already transitioned employees. I will describe my objections and possible strategies in my report.

Data

The introduction, discussion and the theory section speak about that “attitudes evolve” (page 2 introduction) or that “transitioning to permanent employment can alter fairness perceptions” (first paragraph discussion) with many similar phrases throughout. This suggests that the transition between unregular and regular employment is the purpose of the paper. However, the data includes only employees after their transition into regular employment and it also does not entail long-term regular or still-unregular employees to contrast the fairness perceptions between these groups. If these groups exist in the data, I strongly suggest to include them in the analysis to add credit to the data analysis regarding the current story. Without such contrasting, I struggle to see what can be learnt from analysing this specific group of employees because the literature already entails evidence that fairness perceptions are affected when employees feel job insecurity and change when transitioning into and out of employment stages with more and less job security. Bottom line, most of the contributions of the paper described in the discussion section are not analysed in the empirical part of the paper.

Government support

A second major concern relates to the role of government support. Government support is measured on the level of HR management and its consequences on the employee-level. The theoretical arguments why perception of HR managers should affect employees when no direct line management relation exists remains mysterious and is not well grounded in theoretical arguments. Moreover, two more questions arise from the empirical design: (a) why do the government support perceptions moderate the relation between fairness and job perceptions and not between HR-practices and fairness? It sounds much more convincing that HR practices implemented by HR managers who feel a stronger government support for implementing the transition from unregular to regular employment result in higher fairness perceptions of employees than the current hypothesis which sound ad-hoc. (b) I read the questions of the government support more as an understanding of the regulations and opportunities to implement the transition from unregular to regular employment and this understanding further supports the interpretation presented in (a).

Bottom-line regarding the government-support variable, I think modelling the mentioned moderation model would be a great contribution to the literature. I am not aware of any research demonstrating that perceptions of HR managers about support of their work have an influence on employees’ fairness perceptions – but the authors might double-check the claim. Moreover, such a research question also requires exactly the dataset at hand and my comments about the data above are no longer relevant.

Theoretical arguments

The paper argues via the Social Exchange Theory. The current arguments do not address changes in perceptions because of the transition from unregular to regular employment. However, a transition argument is not necessary given the data structure anyhow.

The fairness perceptions cover distributional fairness aspects but not procedural or relational fairness aspects (see Mohrenweiser and Pfeifer 2024, JoHS for a similar design). This is relevant for the theoretical mechanisms. Hence, the theoretical mechanisms should be adjusted to distributional fairness (equity or equality arguments) and the type of fairness should be clearly defined. From a theoretical perspective, we would expect a stronger impact of HR-practices on procedural justice (see the Brockner papers from the 1990’s).

The HR-practices cover only a small area of HR. The choice of HR practices is not well justified – whereby I do not regard the provided reference as a sufficient theoretical justification. The paper could argue that those practices are particularly relevant for employees transitioning from unregular to regular employment. However, it remains unclear why training or flexibility practices are not considered.

Calculating moderation effects

Relying solely on Baron and Kenny for a moderation analysis is a bit dated. I suggest to show the areas of significance using the Neyman and Johnson approach in the moderation figure. I suspect that large parts of relevant data-points show insignificant differences.

Descriptive statistics

The arguments for the data would be much stronger if the paper shows that the demographic characteristics are similar (representatives) for employees in the public sector in South Korea.

Job Satisfaction

I think the relation between fairness to job satisfaction is less interesting. The theoretical arguments regarding the government support on this effect also remain unclear. If the authors want to keep this part, I suggest to include the organisational-level via a fixed effect (see within-between modelling for mixed effect models). Understanding differences within and between organisations might provide an additional edge.

Author Response

(The authors gave the same response as above.)

Round 2

Reviewer 4 Report

Comments and Suggestions for Authors

I thank the authors for their thorough revision and detailed response to my questions. Even if I do not agree with all the responses of the authors and believe that the potential of the paper is not exhausted, I accept the authors’ arguments except one.

My first comment regarding the introduction has not been implemented in the manuscript yet even if the authors agree with my observation. The introduction still claims that the paper analyses the transition between unregular and regular employment. The paper acknowledges that they investigate employees after the transition in the limitations though. Nevertheless, I think that an introduction should introduce what the paper analyses. Hence, I ask the authors to be honest from the start.

Author Response

I attached the response to the comments of reviewer 4. 
